# Non-Angry Superficial Draining Veins: A New Technique in Identifying the Extent of Nidus Excision during Cerebral Arteriovenous Malformation Surgery

**DOI:** 10.3390/brainsci13020366

**Published:** 2023-02-20

**Authors:** Jiandong Zhu, Zhouqing Chen, Weiwei Zhai, Zhong Wang, Jiang Wu, Zhengquan Yu, Gang Chen

**Affiliations:** 1Department of Neurosurgery, The First Affiliated Hospital of Soochow University, 188 Shizi Street, Suzhou 215006, China; 2Department of Neurosurgery & Brain and Nerve Research Laboratory, The First Affiliated Hospital of Soochow University, 188 Shizi Street, Suzhou 215006, China

**Keywords:** indocyanine green video angiography, FLOW 800, cerebral arteriovenous malformation, nidus, draining vein

## Abstract

Background: As essential techniques, intraoperative indocyanine green video angiography (ICG-VA) and FLOW 800 have been widely used in microsurgery for arteriovenous malformations (AVMs). In the present report, we introduced a supplementary technical trick for judging the degree of lesion resection when there were superficial drainage veins. FLOW 800 analysis is used to verify our conjecture. Methods: A retrospective analysis of a 33 case cohort treated surgically from June 2020 to September 2022 was conducted and their lesions were removed by superficial drainage veins as a supplementary technical trick and analyzed with FLOW800. Results: In our 33 AVMs, the feeding artery was visualized earlier than the draining vein. Intraoperatively, the T1/2 peak and slope of the draining vein were significantly higher than that of the lesion. However, the maximum fluorescence intensity (MFI) of the draining vein decreased as the procedure progressed (*p* < 0.001). After reducing the blood flow to the nidus by progressive dissection of the feeding artery, the arteriovenous transit time (AVTT) decreased from 0.64 ± 0.47 s, was prolonged to 2.38 ± 0.52 (*p* < 0.001), and the MFI and slope of the nidus decreased from the pre-resection 435.42 ± 43.90 AI and 139.77 ± 27.55 AI/s, and decreased to 386.70 ± 48.17 AI and 116.12 ± 17.46 AI/s (*p* < 0.001). After resection of the nidus, the T1/2 peak of the draining vein increased from 21.42 ± 4.70 s, prolonged to after dissection of the blood feeding artery, 23.07 ± 5.29 s (*p* = 0.424), and after resection of the lesion, 25.13 ± 5.46 s (*p* = 0.016), with a slope from 135.79 ± 28.17 AI/s increased to 210.86 ± 59.67 AI/s (*p* < 0.001). Conclusions: ICG-VA integrated with FLOW 800 is an available method for determining the velocity of superficial drainage veins. Whether the color of the superficial drainage veins on the cortical surface returns to normal can determine whether the lesion is completely resected and can reduce the possibility of residual postoperative lesions.

## 1. Introduction

Cerebral arteriovenous malformations (AVMs) account for approximately 2% of all hemorrhagic strokes and occur mainly in children and young adults [1,2]. In cases of this disease, the abnormal artery passes through a malformed vascular mass to the vein, and local cerebral hemodynamics are disturbed, resulting in corresponding clinical signs and symptoms [1,3,4]. Although embolization and radiotherapy have made significant progress in recent years, surgical resection remains the most effective treatment for AVMs [5,6,7]. Surgical treatment of complex AVMs still carries a high level of risk, and the strategy and protocol of surgery significantly impact patient prognosis and complications [8]. The feeding arteries, the draining veins, and the malformed vascular mass influence the surgical approach. There are various assessment modalities available, including preoperative magnetic resonance imaging (MRI), computed tomography angiography (CTA), digital subtraction angiography (DSA), intraoperative DSA, intraoperative indocyanine green video angiography (ICG-VA), and Flow 800. The Flow 800 has received much attention as a tool for intraoperative assessment of hemodynamics.

As an essential technique, ICG-VA has been widely used in microsurgery for arteriovenous malformations [9,10,11,12]. ICG-VA provides detailed images of vascular architecture, flow direction, and blood flow delivery, allowing accurate monitoring of hemodynamic changes during all phases of surgical AVM resection. FLOW 800 is a newly developed ICG-VA analysis tool that allows for qualitative visualization of blood flow and vessel intensity. This function color codes images by detailing and can provide time-lapse color maps to immediately identify the direction and sequence of blood flow. Its intensity maps help visualize changes in blood flow over time. Although FLOW 800 has been used in AVM resection, no studies assessing the extent of AVM resection in real-time have been reported, and the use of ICG-VA and FLOW 800 for intraoperative assessment of the scope of AVM lesion resection still lacks attention. The color of the AVM’s drainage vein changes as the procedure progresses. While the accurate assessment of changes in the drainage vein and lesion blood flow information using the FLOW 800 can help the surgeon to develop a surgical strategy for safer, more effective, and complete lesion removal, we believe that hemodynamic changes in AVM draining veins located superficially can serve as one of the markers of complete lesion resection.

This exciting phenomenon inspired us to consider the application of ICG-VA and FLOW 800 to assess the extent of intraoperative lesion resection in AVMs. We validated the value of this technique in our previous clinical practice. This report describes a series of AVMs cases to illustrate this approach and validate its efficacy and safety.

## 2. Materials and Methods

### 2.1. Study Population

We performed a retrospective analysis of our AVMs database, including 54 patients treated with microsurgery from June 2020 to September 2022. We excluded seven patients in whom ICG-VA was not used intraoperatively, five patients in whom DSA was not reviewed postoperatively, five patients in whom surgical data were missing or incomplete, and four patients in whom the lesion was located deep, leaving 33 patients (61.1%) included in the study. All of them used ICG-VA and FLOW 800 intraoperatively. Table 1 briefly summarizes the patient characteristics and surgical outcomes.

All patients underwent MRI or computed tomography (CT) and DSA as routine preoperative examinations to show clear images of the feeding arteries and draining veins, as well as the size of the lesion. Preoperatively, the location and morphology of the lesion and whether it is located in a functional area are clarified based on the MRI findings. Additionally, a thorough preoperative evaluation is performed, for example, whether a complex lesion requires preoperative embolization.

### 2.2. Surgical Technique and Variables

After opening the dura to reveal the lesion and cleaning the surgical area, a new indocyanine green (ICG) solution (25 mg of contrast agent dissolved in 10 mL of sterile saline) was compounded in a single dose of 0.2 to 0.5 mg/kg [10,13]. In the present study, we injected 25 mg of ICG solution into the patient simultaneously. We performed multiple injections as needed, with the total dose injected being approximately 25% of the upper recommended dose of ICG (5 mg/kg), resulting in excellent imaging quality [14]. For ICG angiography, the operator stops operating under the microscope for about 2 min to obtain fluorescent angiographic images. In general, fluorescence angiography is performed in three stages of AVMs excision. First, basic hemodynamic information is received after revealing the malformed vascular mass and its superficial vessels. Second, ICG angiography is performed again after progressive dissection or clamping of the main blood supply artery to obtain changes in the hemodynamic information of AVMs. Finally, ICG imaging was performed at the end of the resection to identify residual AVMs and to label the associated vessels for further analysis. After using fluorescein, the microscope FLOW 800 angiography mode was used to push the contrast agent rapidly. The target area was illuminated with infrared light under the microscope for video capture and observation. All operations were performed using KINEVO 900 (Carl Zeiss Co., Oberkochen, Germany).

The FLOW 800 software integrated into the microscope was used to analyze the reconstructed color visual images of the blood vessels in the surgical area, allowing visualization of the malformed feeding arteries, the draining veins, and the direction and sequence of blood flow. The region of interest (ROI), such as normal veins, draining vein, and cortical arteries, is then selected, and a temporal intensity curve is generated for semiquantitative analysis of the hemodynamic characteristics of the surgical area. At the request of the neurosurgeon, intraoperative fluoroscopic angiography is performed again to determine the extent of lesion resection based on color mapping and time–intensity curves. After complete resection of the lesion, fluorescence angiography was performed to show that the lesion and the blood supply artery were not visible and the draining veins were patent, which means that the lesion was completely resected. Related parameters were defined as follows: time to half-maximal fluorescence was defined as the time required for the ICG fluorescence intensity to reach 50% of the maximum value (T1/2 peak); transit time was defined as the time needed for blood to flow from the artery to vein (arteriovenous transit time, AVTT); rise time was defined as the interval between 10% and 90% of the maximum signal; and cerebral blood flow index slope was defined as the ratio of maximum fluorescence intensity (MFI) to rise time.

We carefully collected and recorded primary demographic data and imaging presentations for each patient, including clinical presentation, Spetzler–Martin grade, anatomic location of the lesion, intraoperative conditions, and FLOW 800 results, and compiled preoperative and postoperative follow-up modified Rankin scales.

### 2.3. Statistical Analysis

Values are presented as the mean ± standard deviation and ordinal values are presented as median and quartile deviation. The statistical analysis was performed and visualized using R (3.6.3) software and the “ggplot2” R package. After confirming the normal distribution pattern using the Shapiro–Wilk normality test and the variance distribution using Levene’s test, differences between groups were assessed using the Kruskal–Wallis test or *T*-test to analyze the intraoperative ICG parameters, and the comparison between groups was performed using Dunn’s test. Analysis of prognostic parameters was performed using Wilcoxon rank sum test. A *p* value < 0.05 was considered statistically significant.

## 3. Results

### 3.1. Clinical Profile

Our study included 33 patients with different clinical presentations, with a mean age of 35.4 years, and more males than females (24 males vs. 9 female patients). Among our 33 patients, 9 lesions were located in the frontal lobe (27.3%), 5 in the temporal lobe (15.2%), 4 in the parietooccipital lobe (12.1%), 4 in the parietal lobe (12.1%), 4 in the occipital lobe (12.1%), 3 in the cerebellum (9.1%), 2 in the frontoparietal lobe (6.1%), and 2 in the temporooccipital lobe (6.1%). Twelve patients had grade 1 Spetzler–Martin AVM, twelve had grade 2 AVM, and nine had grade 3 AVM. In total, 48.5% of the patients presented clinically with headache, 18.1% with dizziness, 15.1% with epilepsy, 15.1% with impaired consciousness, and one with incidental findings (Table 1). Intracranial arteriovenous malformations were found on preoperative DSA in all patients. In one patient (Case 28), DSA was not performed due to preoperative urgency, and preoperative CT angiography suggested a tortuous and thickened intracranial vascular mass (Figure 1).

### 3.2. Intra-Operative Fluorescein Angiography

Both ICG-VA and FLOW 800 integration have been successfully performed in the surgical microscope, allowing a clear distinction between feeding arteries and draining veins in superficial vessels or exposed deep vessels, as well as hemodynamic changes during surgery, based on the visualization and analysis of the results (Figure 2 and Figure 3). In total, 109 ICG-VA procedures were performed in 33 patients and 54 feeding arteries were identified. On average, 3.3 angiograms were performed per patient (ranging from 3 to 4), 1–4 feeding arteries were identified intraoperatively in 33 patients, and 31 patients also had superficial draining veins (Figure 1, Figure 2 and Figure 3). After clipping the feeders, ICG-VA revealed that the flow of the draining vein was reduced. After total dissection of the nidus, ICG-VA showed thickened ICG filling in the superficial draining vein (Figure 2 and Figure 3). Flow800 showed a gradually slowed down flow in the draining vein by a delay map. After dissection of the nidus, a delayed image of the draining vein perfused by clipped feeders looked dark green, meaning the flow of the draining vein was reduced and slowed. Finally, after total dissection of the nidus, almost all of the superficial draining veins looked blue (Figure 1, Figure 2 and Figure 3). All draining veins showed standard venous flow patterns at the end of the procedure, with no arterial flow disturbances.

In our 33 AVMs, the feeding artery was visualized earlier (T1/2 peak 22.46 ± 5.64 s) than the draining vein (T1/2 peak 21.42 ± 4.70 s) and could be used as a basis for identifying the feeding artery (Figure 4B). Intraoperatively, the T1/2 peak and slope of the draining vein were significantly higher than that of the lesion (Figure 4E,F). However, the MFI of the draining vein decreased from 388.16 ± 48.83 AI to 218.52 ± 49.62 AI (*p* < 0.001) as the procedure progressed (Figure 5D). After reducing the blood flow to the nidus by progressive dissection of the feeding artery, the AVTT decreased from 0.64 ± 0.47 s and was prolonged to 2.38 ± 0.52(*p* < 0.001) (Figure 4J), the MFI decreased from 435.42 ± 43.90 AI to 386.70 ± 48.17 AI (*p* < 0.001) (Figure 5A), and the slope of the nidus decreased from the pre-resection 139.77 ± 27.55 AI/s to 116.12 ± 17.46 AI/s (*p* < 0.001) (Figure 5C). After resection of the nidus, the arteriovenous shunt disappeared and the T1/2 peak of the draining vein increased from 21.42 ± 4.70 s, prolonged to 25.13 ± 5.46 s (*p* = 0.016) (Figure 5E), with a slope from 135.79 ± 28.17 AI/s, increased to 210.86 ± 59.67 AI/s (*p* < 0.001) (Figure 5F). At this time, the blood flow in the draining and normal veins was essentially equal, with MFIs of 218.52 ± 49.62 AI and 245.27 ± 36.73 AI, respectively (*p* = 0.036)(Figure 4G).

### 3.3. Prognosis and Follow-Up Results

DSA was performed postoperatively in all cases and no residual lesions were found. No significant postoperative complications were observed in all patients. After postoperative follow-up (one patient was lost, 96.8% follow-up), the patients’ modified Rankin scale improved from 2 (1–3) preoperatively to 1 (0–1.25) at follow-up (*p* = 0.002).

## 4. Discussion

The most important feature of successful AVM microsurgery is that the draining vein must be preserved until the last moment of lesion excision [8]. The main reason for this is to avoid premature blockage of the draining vein, leading to increased pressure within the malformed vascular mass and subsequent rupture, leading to uncontrolled bleeding. In addition, the draining vein helps the neurosurgeon guide the dissection of the lesion. One third of the malformations are located below the cortical surface, and 2/3 have superficial venous drainage [5]. This surgical approach and these anatomic features also support the feasibility of the specialized technique we describe. Determining superficial drainage vein color is most accessible and most straightforward when superficial drainage veins are present. Of the 33 cases reported here, 31 were found to have draining veins on the cortical surface. The malformed vascular mass was visible at depth depending on where the draining veins were located on exploration. In the context of AVM microsurgery applications, Fukuda et al. demonstrated the effectiveness of FLOW 800 for understanding hemodynamic changes at various stages of AVM surgical resection [14]. Kato obtained similar conclusions in a retrospective analysis of 17 patients who underwent surgical resection of intracranial AVMs [15]. We compared the T1/2 peak, slope, and maximum fluorescence intensity of the nidus and the draining vein. We showed that the T1/2 peak of both the nidus and the draining vein was earlier before and after resection. The slope of the nidus was significantly less than before resection, the slope of the draining vein was substantially greater after surgery than before, and the maximum fluorescence intensity was maximum before resection. As the artery feeding the lesion was gradually disconnected, the arteriovenous transit time was progressively prolonged. We are concerned that with the intraoperative separation and severance of the blood supply artery, blood flow in the malformed vascular mass decreases, local circulation times become more protracted, average arteriovenous return is restored, and venous blood flow increases. This phenomenon is also well explained by the fact that the MFI of the draining vein and the usual vein is essentially the same after complete resection of the lesion. Our measurements are consistent with the results of Ye and Ng. [16,17], but Kato et al. showed that all semiquantitative measures were similar between the feeders, niduses, and drainers [15]. This may be due to differences in the size of the niduses in the study population and the abnormally high flow in the draining veins supplied directly through the lesion via the niduses. When residual lesions are suspected, they can be evaluated in conjunction with preoperative imaging and reevaluated based on hemodynamic changes during progressive dissection of the supply artery. Multiple intraoperative ICG injections are allowed, and a reduction in maximum flow was achieved in this study. ICG-VA imaging is influenced by the dose used. Some groups have used a dose of 0.1 mg/kg for a single injection [18]. However, the efficacy and safety of this phenomenon were not evaluated, and we injected 25 mg of ICG solution into the patient and obtained excellent imaging results. Conventional ICG-VA can help the surgeon instantly identify the direction and recovery of blood flow in AVMs; however, these quantitative assessments with FLOW 800 provide precise information about changes in lesion and vessel blood flow, primarily since it provides instant color images that are of tremendous significance in determining the draining vein and the supplying artery. Although the surgeon may be more focused on the operation itself during the procedure, these assistive technologies can significantly help the neurosurgeon when situations arise; the naked eye cannot determine that.

We believe that intraoperative hemodynamic monitoring of the superficial drainage veins using ICG-VA and FLOW 800 can help us to identify the extent of complete resection of the lesion, avoid residual deformities, and reduce the possibility of severe consequences such as postoperative rebleeding. We have found that changes in the color of the superficial drainage veins indicate changes in the hemodynamics of the blood in the vessel and that intraoperative changes in the color of the superficial drainage veins can help the surgeon determine the extent of lesion removal (Figure 6). We present our experience using ICG-VA and FLOW 800 in the AVMs to observe the drainage veins to identify the extent of lesion resection. No residual lesions were found in any of the 33 patients who underwent DSA at postoperative follow-up. Taddei and Takagi came to the same conclusions and even suggested that ICG-VA could be an alternative to intraoperative or postoperative DSA [11,19]. After analyzing FLOW 800 fluorescence angiographic data, Feng et al. concluded that ICG-VA effectively revealed the vascular configuration of superficial arteriovenous malformations and that a 100% total resection rate could be achieved with ICG-VA guidance [14]. Hänggi et al. reported 15 patients with cerebral arteriovenous malformations in whom ICG-VA was performed intraoperatively [20]. They concluded that the most common procedure-related complications were an injury to routine cerebral arterial branches and incomplete resection of the lesion. It exemplifies the critical role of ICG-VA hemodynamic analysis of intraoperative drainage veins in the intraoperative identification and protection of regular cerebrovascular branches. The absence of new neurological deficits in any of the patients in our study postoperatively also supports this view.

With the use of ICG-VA and FLOW 800, it can help us to manage AVMs during surgery better. However, there are still some exceptional cases, for example, when the patient is too ill to complete FLOW 800, when AVMs are found by chance on intraoperative exploration, or when there is unpredictable hemorrhage intraoperatively. The extent of lesion resection cannot be judged. Based on previous experience, many neurosurgeons often become overwhelmed in these cases, and the technique we describe can be a useful additional tool for determining the extent of AVM resection. The combined application of this method, combined with the intraoperative surgeon’s experience, can help the neurosurgeon more wholly and effectively resect the lesion while simultaneously avoiding unnecessary exploration and minimizing damage to the brain parenchyma. At the same time, it enables comprehensive control of the extent of resection during the procedure and increases the surgeon’s self-confidence. Our study validated the safety and efficacy of this technique by FLOW 800 statistical analysis, with no new neurological deficits after surgery in all cases. All patients showed significant improvement at follow-up compared to preoperative. The main reason is the lack of capillaries between the arteries and veins during embryonic development, which allows direct connection between high-flow arterioles and low-resistance veins. This phenomenon enables intra-arterial blood to enter the veins directly. It provides blood around the lesion, in order to “steal” blood from surrounding brain tissue by shunting it more easily through these low resistance “bridges” rather than the surrounding capillary network. Thus, many AVMs tend to have arterialized veins, and this idea was confirmed by the immunohistochemical study of Hoya et al. [21]. Our study used the intraoperative color change of arterialized veins as an intraoperative marker for lesion resection. It was validated using FLOW 800, demonstrating the hemodynamic changes in the draining veins during lesion resection. Although our patients had good results with intraoperative ICG-VA, the lesions were superficial in our cases, and deep lesions, such as those in the ventricles or thalamus were not included in this study. Two of the 33 patients in our study had no intraoperative findings of draining veins located on the cortical surface. This may be because the fact that the draining veins are located deep in the nidus, which cannot be revealed intraoperatively due to the limitations of ICG-VA, and the fact that the draining veins often contain mixed blood, making it difficult to accurately identify the arteries and veins under the microscope [22].

It is worth emphasizing that the surgical technique we reported can help surgeons to assess the extent of intraoperative resection of arteriovenous malformations and improve the safety and efficacy of the procedure. However, as we described previously, due to the inherent limitations of ICG-VA, it can only show superficial vascular structures. It cannot clearly show deep anatomy, especially when obscured by normal brain tissue or malformed lesions. Therefore, we excluded four cases in which the lesions were deep (including AVMs in the ventricles). Although ICG-VA cannot visualize deep vessels, based on our intraoperative experience, this technique is still of tremendous help to surgeons in determining the extent of resection of arteriovenous malformations located deep in the brain, which have branch vessels returning to the superficial veins. In addition, when AVMs rupture and bleed, different bleeding patterns may lead to different clinical outcomes [23], such as arachnoid hemorrhage and intraventricular hemorrhage, and additional bleeding can cause hemodynamic disturbances in the original lesion, adding to the challenge of surgical resection of arteriovenous malformations. From the case we described of brain parenchymal hemorrhage with rupture into the lateral ventricle, this technique is also applicable to this situation, allowing a more rapid and accurate determination of the extent of lesion resection. However, more prospective studies are needed to confirm this.

However, there are some limitations to this study. Firstly, only 33 patients were included in the study, and there is a risk that the reliability of the results may be reduced due to insufficient sample size when comparing data from multiple groups. Secondly, our study did not use other intraoperative blood flow evaluation tools to compare the results with FLOW 800 color fluoroscopy. In addition, several variables, such as blood pressure, heart rate, and ICG dye injection rate, can affect the ICG-VA hemodynamic analysis curve. These minor differences can be amplified as deviations in hemodynamic parameters. Finally, the nature of the study precluded our retrospective analysis of more meaningful FLOW 800 parameters, such as changes in hemodynamics after hematoma clearance or intraoperative bleeding.

## 5. Conclusions

The ICG-VA combined with the FLOW 800 can safely, quickly, and reliably identify the vascular structures of superficial AVMs. Whether the color of the superficial drainage veins on the cortical surface returns to normal can determine whether the lesion is completely resected and can reduce the possibility of residual postoperative lesions. Additionally, as a surgical technique, it can help neurosurgeons cope with some unpredictable intraoperative situations and increase the surgeon’s self-confidence.

## Figures and Tables

**Figure 1 brainsci-13-00366-f001:**
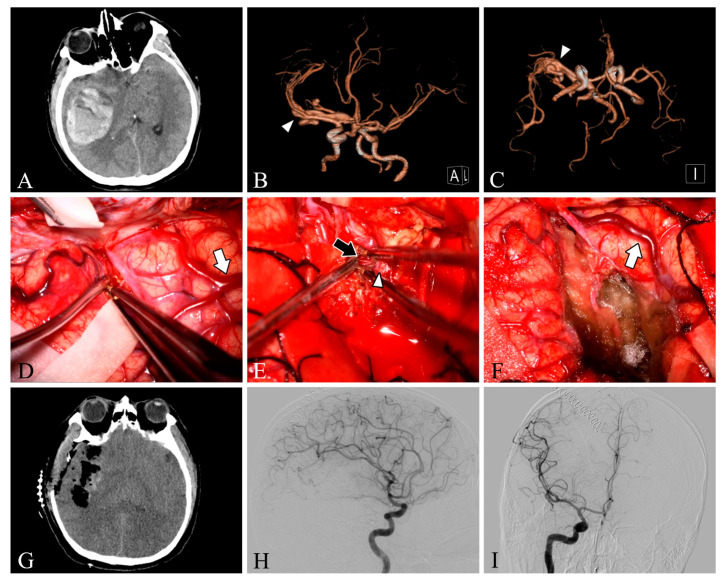
Case 28. A patient with an arteriovenous malformation and intracranial hemorrhage in the right temporal lobe underwent surgery. Preoperative axial CT image (**A**) showing cerebral hemorrhage in the right temporal lobe. Preoperative CT angiography (**B**,**C**) demonstrates a tortuous and thickened vascular mass (white arrowheads) in the M2 segment of the right middle cerebral artery. After resection, the lesion was removed and the superficial draining vein (white arrows) changed from bright red to dark red (**D**,**F**). The feeding artery was shown intraoperatively (**E**, black arrow). Postoperative axial CT image (**G**) showed that the hematoma was cleared entirely. Postoperative lateral (**H**) and anteroposterior (**I**) views of internal carotid artery angiography showed no residual arteriovenous malformation.

**Figure 2 brainsci-13-00366-f002:**
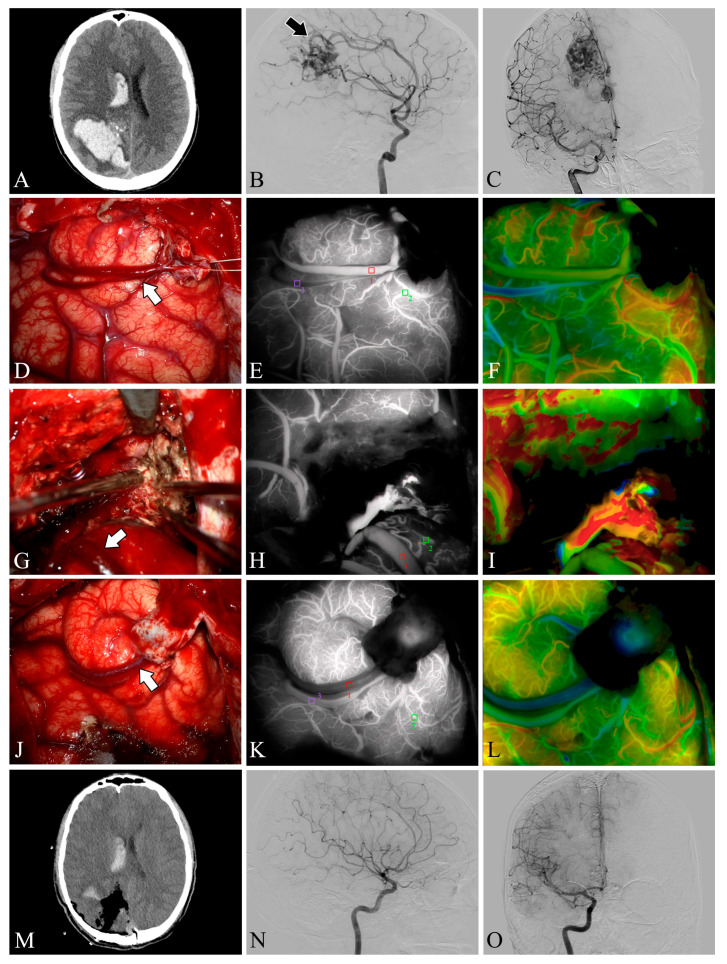
Case 5. A patient with an arteriovenous malformation and intracranial hemorrhage in the right parietooccipital lobe underwent surgery. Preoperative axial CT image (**A**) showing cerebral hemorrhage in the right parietooccipital lobe and breaking into the ventricle. Preoperative lateral (**B**) and anteroposterior (**C**) views of internal carotid artery angiography, the nidus fed by the anterior cerebral artery can be seen (black arrow). The lesion was removed and the superficial draining vein (white arrows) changed from bright red to dark red after resection (**D**,**G**,**J**). Flow intensity analysis during surgery using indocyanine green fluorescent video angiography (**E**,**H**,**K**) and FLOW 800 (**F**,**I**,**L**). Postoperative axial CT image (**M**) showed that the hematoma was cleared entirely. Postoperative internal carotid artery angiography (**N**,**O**) showed no residual arteriovenous malformation.

**Figure 3 brainsci-13-00366-f003:**
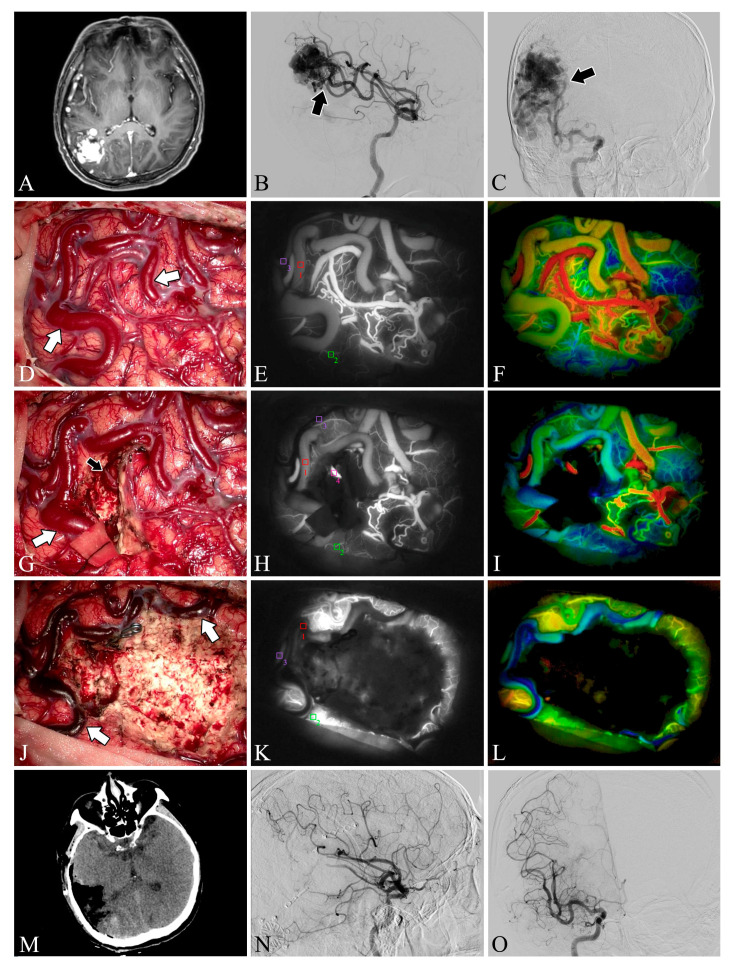
Case 22. A patient with an arteriovenous malformation in the right parietooccipital lobe underwent surgery. Preoperative axial MRI image (**A**) showing the abnormal hyperintense mass in the right parietooccipital lobe. Preoperative lateral (**B**) and anteroposterior (**C**) views of internal carotid artery angiography. A Spetzler–Martin grade III AVM fed by the middle cerebral artery (black arrow) can be seen in the right parietooccipital lobe. The lesion was removed and the blood flow in the superficial draining vein (white arrows) decreased (**D**,**G**,**J**). Flow intensity analysis during surgery using indocyanine green fluorescent video angiography (**E**,**H**,**K**) and FLOW 800 (**F**,**I**,**L**). Postoperative axial CT image (**M**) showed that the lesion was cleared. Postoperative internal carotid artery angiography (**N**,**O**) showed no residual arteriovenous malformation.

**Figure 4 brainsci-13-00366-f004:**
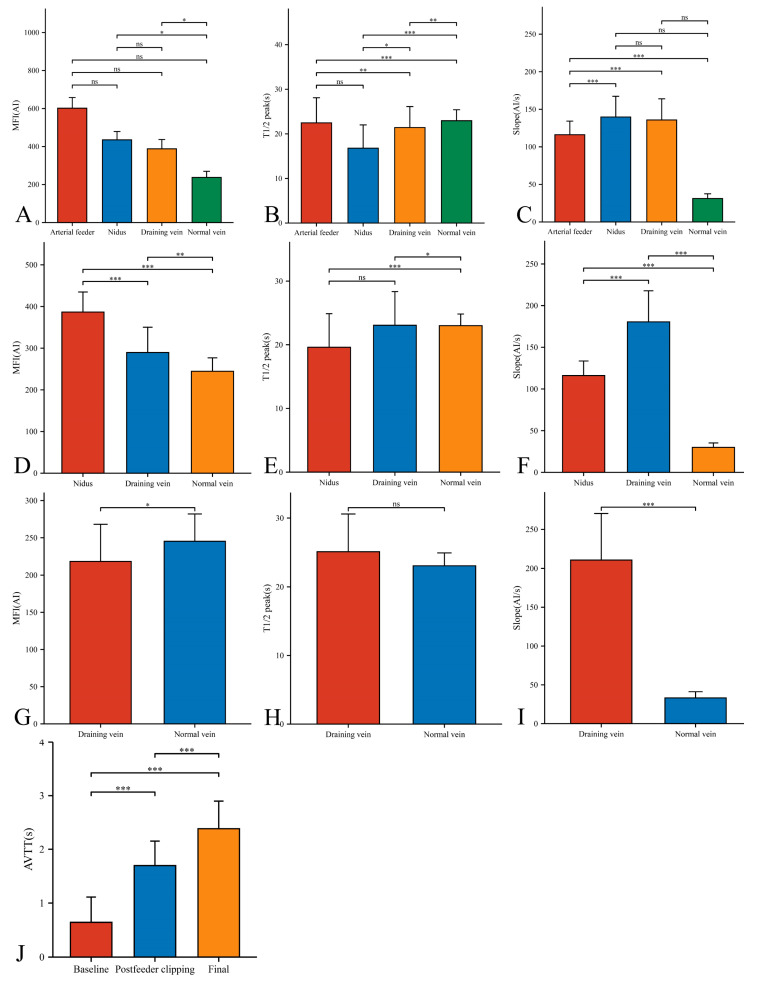
Hemodynamic changes of arterial feeder, nidus, draining vein, and normal vein in the timing of baseline phase (**A**–**C**), postfeeder clipping phase (**D**–**F**), and final phase (**G**–**I**). AVTT in the timing of baseline, postfeeder clipping, and final phase (**J**) (*, *p* < 0.05; **, *p* < 0.01; ***, *p* < 0.001; ns, *p* ≥ 0.05). MFI, maximum fluorescence intensity; AI, arbitrary intensity; T1/2 peak, time to half-maximal fluorescence; Slope, slope of rise; AVTT, arteriovenous transit time.

**Figure 5 brainsci-13-00366-f005:**
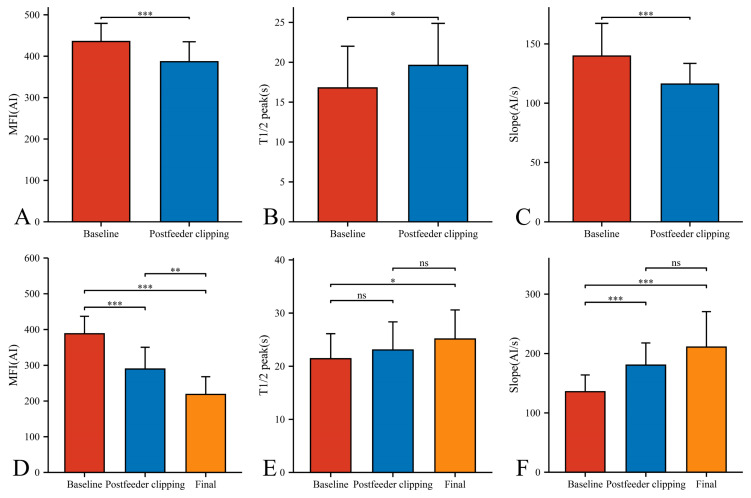
Hemodynamic changes of nidus (**A**–**C**) and draining vein (**D**–**F**) in the timing of baseline, postfeeder clipping, and final phase (*, *p* < 0.05; **, *p* < 0.01; ***, *p* < 0.001; ns, *p* ≥ 0.05). MFI, maximum fluorescence intensity; AI, arbitrary intensity; T1/2 peak, time to half-maximal fluorescence; Slope, slope of rise.

**Figure 6 brainsci-13-00366-f006:**
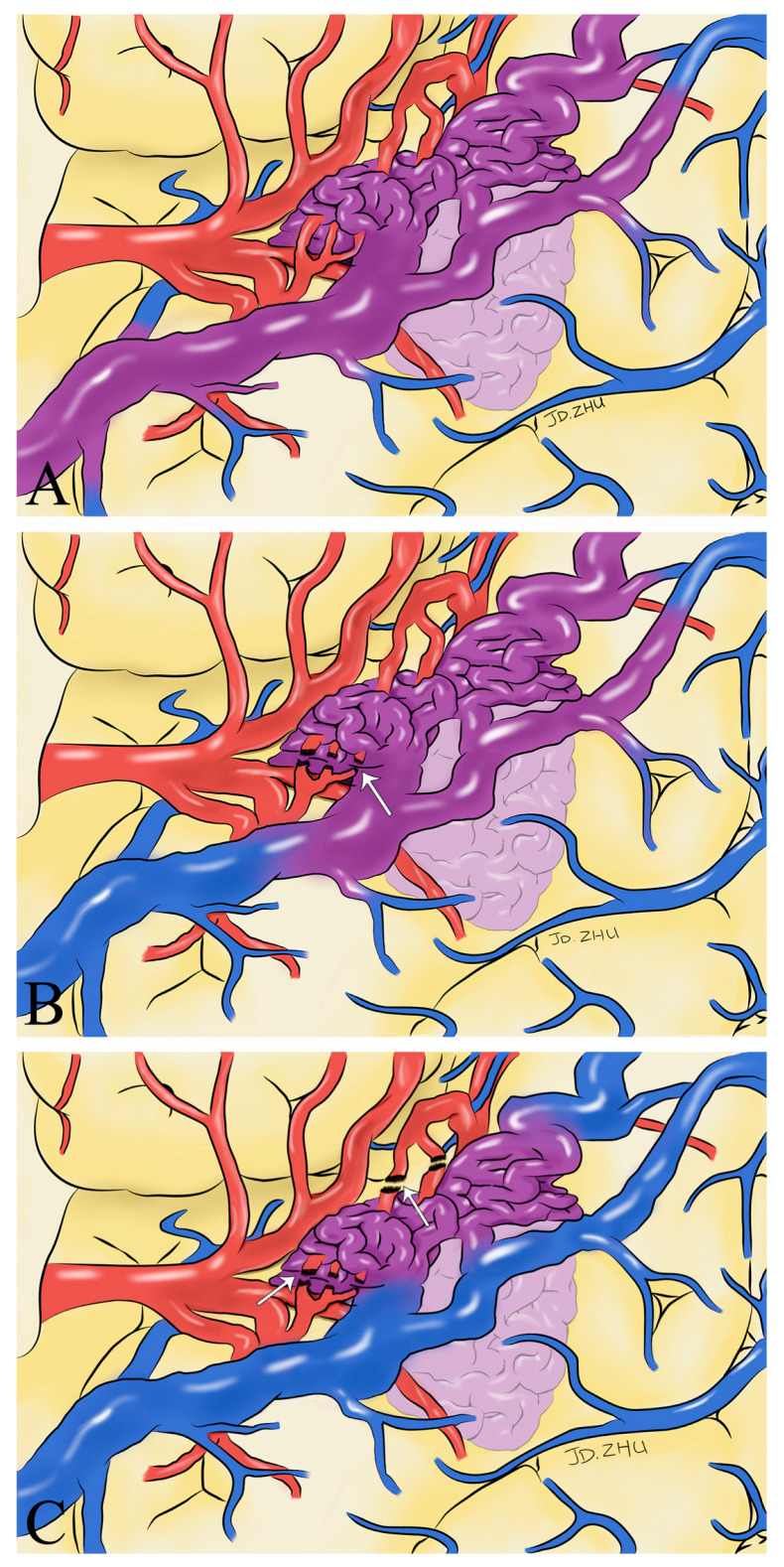
Schematic drawings show hemodynamic changes during AVM resection. Before the feeding artery was clipped, the blood in the superficial draining vein was mixed blood of arteries and veins (purplish red) (**A**). With the detachment of the feeding artery (white arrow), the superficial draining vein gradually returned to normal hemodynamics (**B**). After the complete detachment of the feeding artery (white arrow), normal venous blood returned to the draining vein (blue) (**C**).

**Table 1 brainsci-13-00366-t001:** Characteristics of patients with AVMs and Surgical Outcomes *.

						Visualization			mRs Score	
Case NO.	Age, Sex	S-M	Location (Nidus)	Presentation	Size (mm)	NO. of Feeders	Nidus	Drainer	Pre- Operative	Follow- Up
1	21, M	2	Parietooccipital	Headache	21 × 56 × 40	4	Yes	Yes	1	0
2	19, M	3	Occipital	Headache	41 × 35 × 45	2	Yes	Yes	1	1
3	37, M	2	Frontal	Headache	15 × 14 × 33	2	Yes	Yes	3	1
4	26, M	1	Cerebellum	Headache	11 × 8 × 9	1	Yes	Yes	2	0
5	14, M	2	Parietooccipital	Headache	28 × 36 × 30	1	Yes	Yes	4	2
6	48, M	2	Frontal	Unconsciousness	27 × 20 × 15	1	Yes	Yes	5	3
7	27, F	3	Frontal	Epilepsy	21 × 35 × 25	1	Yes	Yes	1	1
8	31, F	1	Parietal	Incidental	20 × 23 × 20	1	Yes	Yes	0	0
9	24, M	3	Frontal	Epilepsy	31 × 29 × 35	1	Yes	Yes	1	0
10	49, M	1	Occipital	Headache	18 × 14 × 15	1	Yes	Yes	2	2
11	46, M	1	Occipital	Unconsciousness	21 × 22 × 20	1	Yes	Yes	5	3
12	46, M	3	Temporal	Epilepsy	34 × 28 × 20	2	Yes	Yes	1	0
13	64, M	3	Frontoparietal	Dizzy	15 × 55 × 30	2	Yes	Yes	2	NA
14	60, M	3	Temporooccipital	Headache	38 × 24 × 25	1	Yes	Yes	1	1
15	42, M	2	Temporal	Dizzy	40 × 41 × 25	1	Yes	Yes	1	0
16	52, M	2	Temporooccipital	Headache	13 × 14 × 35	1	Yes	Yes	3	1
17	37, F	2	Frontal	Unconsciousness	23 × 31 × 20	1	Yes	Yes	5	3
18	53, F	1	Occipital	Dizzy	28 × 20 × 15	1	Yes	Yes	2	1
19	39, M	1	Cerebellum	Headache	9 × 25 × 5	1	Yes	Yes	2	0
20	31, M	1	Parietooccipital	Unconsciousness	19 × 14 × 28	2	Yes	Yes	5	4
21	37, F	2	Frontal	Headache	32 × 19 × 30	2	Yes	Yes	1	1
22	49, M	3	Parietooccipital	Headache	36 × 49 × 30	3	Yes	Yes	2	1
23	42, F	3	Parietal	Headache	50 × 32 × 40	2	Yes	Yes	3	2
24	24, F	2	Frontoparietal	Headache	5 × 14 × 7	1	Yes	Yes	3	1
25	23, M	1	Parietal	Dizzy	6 × 7 × 4	2	Yes	Yes	2	0
26	13, M	3	Frontal	Epilepsy	49 × 30 × 22	1	Yes	Yes	1	0
27	29, M	1	Temporal	Headache	22 × 17 × 18	4	Yes	Yes	1	1
28	33, M	1	Temporal	Unconsciousness	20 × 9 × 15	1	Yes	Yes	5	4
29	29, M	1	Cerebellum	Headache	11 × 13 × 18	1	Yes	Yes	2	1
30	41, F	2	Parietal	Dizzy	24 × 10 × 30	2	Yes	Yes	1	0
31	49, M	2	Temporal	Dizzy	38 × 33 × 30	2	Yes	Yes	1	1
32	33, M	3	Frontal	Epilepsy	21 × 32 × 20	3	Yes	No	0	0
33	29, F	1	Parietooccipital	Headache	22 × 25 × 25	1	Yes	No	3	1

* S-M, Spetzler–Martin Grade; DSA, digital subtraction angiography; mRS, modified Rankin scale; M, male; F, female; NA, not available.

## Data Availability

The data supporting this study’s findings are available from the corresponding author, J.W., upon reasonable request. Due to the sensitivity of the data used in this study, the data are not publicly available since this could compromise the privacy of research participants.

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
