# Peer review of "Non-Angry Superficial Draining Veins: A New Technique in Identifying the Extent of Nidus Excision during Cerebral Arteriovenous Malformation Surgery"

_brainsci, 2023, doi:10.3390/brainsci13020366_

Round 1

Reviewer 1 Report

We read with interest the study entitled: "Non-angry superficial draining veins: a new technique in identifying the extent of nidus excision during cerebral arteriovenous malformation surgery". 

The study is well conceived and describe an interesting technique to obtain precise and complete removal of an AVM. 

I would suggest to better underline potential limits of this technique, especially for neurosurgeons not experienced in AVM vascular surgery. Is the technique safe and feasible even for deeply located AVMs, or for AVM with multiple feeders/drainages? Can you please briefly comment on these aspects? 

Another interesting point to underline is the, in the event of a haemorrhage, is the relevance of bleeding pattern on clinical appearance and outcome  (eg Sturiale CL et al Relevance of bleeding pattern on clinical appearance and outcome in patients with hemorrhagic brain arteriovenous malformations. J Neurol Sci. 2013 Jan 15;324(1-2):118-23. doi: 10.1016/j.jns.2012.10.016. Epub 2012 Nov 9. PMID: 23146614. ). You could add some lines on that in discussion, in order to implement this type of surgery in some specific type of AVMs.

Author Response

Point 1: I would suggest to better underline potential limits of this technique, especially for neurosurgeons not experienced in AVM vascular surgery. Is the technique safe and feasible even for deeply located AVMs, or for AVM with multiple feeders/drainages? Can you please briefly comment on these aspects?

Response 1: We have made corrections according to the Reviewer’s comments. In response to the potential limitations of this technology, we highlight it in the DISCUSSION. The technique we describe is still applicable in our surgical experience, deep AVMs in the presence of superficial draining veins. Our cases included 14 patients with AVMs with multiple feeders (we summarized and showed them in the manuscript [table1 and figure 3]), so the technique is applicable in this scenario. As this study was a retrospective analysis, AVMs with multiple drainages were not identified in our cases, and we will continue to draw lessons from our experience in future work.

Point 2: Another interesting point to underline is the, in the event of a haemorrhage, is the relevance of bleeding pattern on clinical appearance and outcome  (eg Sturiale CL et al Relevance of bleeding pattern on clinical appearance and outcome in patients with hemorrhagic brain arteriovenous malformations. J Neurol Sci. 2013 Jan 15;324(1-2):118-23. doi: 10.1016/j.jns.2012.10.016. Epub 2012 Nov 9. PMID: 23146614. ). You could add some lines on that in discussion, in order to implement this type of surgery in some specific type of AVMs.

Response 2: It is really true as Reviewer suggested that bleeding pattern correlate with clinical outcomes. In conjunction with our case, we added this section to the discussion.

These sections are all added in the penultimate paragraph of the DISCUSSION.

Reviewer 2 Report

The authors have submitted a manuscript describing their use of ICG fluorescence and 800 flow integration in assessing draining veins during AVM resection. The manuscript is well written. It describes a potentially useful adjunct to AVM resection.

A few suggestions to improve the manuscript follow:

1. In line 91 in METHODS, the authors reference a dose of ICG that others have used. This statement is not part of the authors METHODS, so it should be removed from METHODS. The authors could comment about pros and cons of this dose in DISCUSSION.

2. In METHODS, the authors should address whether or not normal adjacent veins and arteries were visualized with ICG fluorescence and Flow 800 integration and measured for comparison? If not, this deficit represents a limitation of the manuscript.

3. In RESULTS, the authors should address if adjacent hemorrhage or elevated intracranial pressure alters measurements, and if measurements change after evacuation of hemorrhage or reduction of intracranial pressure. If this data was not collected, then this deficit also represents a limitation.

4. In RESULTS, the authors should describe how measurements change after control of intraoperative bleeding from AVM. If this data was not collected, the deficit should be discussed in limitations.

5. In FIGURE 6. panels A and B look similar, except for the coloring of the draining vein. Panel C shows cuts in the feeding arteries, but the nidus still appears present. Therefore, the legend does not appear to correspond to the panels, or the changes between panels are not obvious enough to make the authors’ points. The authors should use arrow labels, or they should make more obvious changes between panels.

Author Response

Point 1: In line 91 in METHODS, the authors reference a dose of ICG that others have used. This statement is not part of the authors METHODS, so it should be removed from METHODS. The authors could comment about pros and cons of this dose in DISCUSSION.

Response 1: As Reviewer suggested that we have removed the reference in the METHODS and added this section to the DISCUSSION(Line 265 ).

Point 2: In METHODS, the authors should address whether or not normal adjacent veins and arteries were visualized with ICG fluorescence and Flow 800 integration and measured for comparison? If not, this deficit represents a limitation of the manuscript.

Response 2: Due to the limitations of the retrospective study, we performed FLOW800 analysis only on normal adjacent veins. We have presented the results in Figure 4 (A-F) with additional notes in the METHODS. Unfortunately, we did not analyze the normal adjacent arteries, which we have added as a limitation of the manuscript.

Point 3 and Point 4:

3: In RESULTS, the authors should address if adjacent hemorrhage or elevated intracranial pressure alters measurements, and if measurements change after evacuation of hemorrhage or reduction of intracranial pressure. If this data was not collected, then this deficit also represents a limitation.

4: In RESULTS, the authors should describe how measurements change after control of intraoperative bleeding from AVM. If this data was not collected, the deficit should be discussed in limitations.

Response 3 and 4: The nature of the study prevented additional monitoring of this component of our research, and we have added to the limitations of the manuscript.

Point 5: In FIGURE 6. panels A and B look similar, except for the coloring of the draining vein. Panel C shows cuts in the feeding arteries, but the nidus still appears present. Therefore, the legend does not appear to correspond to the panels, or the changes between panels are not obvious enough to make the authors’ points. The authors should use arrow labels, or they should make more obvious changes between panels.

Response 5: We apologize for the confusion our diagram has caused you. Figure 6 is used to illustrate the problem that the color of the superficial draining vein changes from purplish red at the beginning (mixed arterial blood) to blue at the end (venous blood) as the donor artery is disconnected, thus helping the readers to understand better the mechanism of the color change of the vessel in the surgical view. We have modified the legend and added arrow labels to better illustrate this issue.

Reviewer 3 Report

The authors reported the application of FLOW 800 for a supplementary technical trick for judging the degree of lesion resection when there were superficial drainage veins. ICG-VA has been widely used in microsurgery for AVM and provides detailed images of vascular architecture, flow direction, and blood flow delivery, allowing accurate monitoring of hemodynamic changes during all phases of surgical AVMs resection. FLOW 800 is a newly developed ICG-VA analysis tool that allows for qualitative visualization of blood flow and vessel intensity.  I appreciated the article and the application of FLOW 800 for determining the velocity of superficial drainage veins in AVM surgery. On the other han, the manuscripts has some limitations that the authors reported properly.

Author Response

We would like to express our great appreciation to Reviewer for comments on our paper.